# Transcriptome Analysis Reveals the Stress Tolerance to and Accumulation Mechanisms of Cadmium in *Paspalum vaginatum* Swartz

**DOI:** 10.3390/plants11162078

**Published:** 2022-08-09

**Authors:** Lei Xu, Yuying Zheng, Qing Yu, Jun Liu, Zhimin Yang, Yu Chen

**Affiliations:** College of Agro-Grassland Science, Nanjing Agricultural University, Nanjing 210095, China

**Keywords:** Cadmium, *Paspalum vaginatum*, RNA-seq, qRT-PCR, *PvSnRK2.7*

## Abstract

Cadmium (Cd) is a non-essential heavy metal and high concentrations in plants causes toxicity of their edible parts and acts as a carcinogen to humans and animals. *Paspalum vaginatum* is widely cultivating as turfgrass due to its higher abiotic stress tolerance ability. However, there is no clear evidence to elucidate the mechanism for heavy metal tolerance, including Cd. In this study, an RNA sequencing technique was employed to investigate the key genes associated with Cd stress tolerance and accumulation in *P. vaginatum*. The results revealed that antioxidant enzyme activities catalase (CAT), peroxidase (POD), superoxide dismutase (SOD), and glutathione S-transferase GST) were significantly higher at 24 h than in other treatments. A total of 6820 (4457/2363, up-/down-regulated), 14,038 (9894/4144, up-/down-regulated) and 17,327 (7956/9371, up-/down-regulated) differentially expressed genes (DEGs) between the Cd1 vs. Cd0, Cd4 vs. Cd0, and Cd24 vs. Cd0, respectively, were identified. The GO analysis and the KEGG pathway enrichment analysis showed that DEGs participated in many significant pathways in response to Cd stress. The response to abiotic stimulus, the metal transport mechanism, glutathione metabolism, and the consistency of transcription factor activity were among the most enriched pathways. The validation of gene expression by qRT-PCR results showed that heavy metal transporters and signaling response genes were significantly enriched with increasing sampling intervals, presenting consistency to the transcriptome data. Furthermore, over-expression of *PvSnRK2.7* can positively regulate Cd-tolerance in *Arabidopsis*. In conclusion, our results provided a novel molecular mechanism of the Cd stress tolerance of *P. vaginatum* and will lay the foundation for target breeding of Cd tolerance in turfgrass.

## 1. Introduction

Heavy metal pollution is a major environmental concern that severely damages plant, animal, and human health [1]. Cadmium (Cd) is one of the non-essential heavy metals and one of the most toxic pollutants, deleterious to the environment and agriculture [2]. The higher accumulation of Cd in the edible parts of plants that enter the human and animal food chain thus leads to severe health risks [3]. Inorganic fertilizer application and spraying of synthetic fungicides are the major sources of Cd contamination in the food chain [4]. In addition, a higher accumulation of Cd in plants reduces seed germination, early seedling growth, plant biomass, physiological, and biochemical processes of the plant such as photosynthetic gas exchange parameters, soluble sugar, and soluble protein and chlorophyll synthesis in plants [5]. Additionally, Cd concentration in plants activates reactive oxygen species such as hydrogen peroxide (H_2_O_2_), singlet oxygen (O_2_), organic hydroperoxide (ROOH), and oxygen-derived free radicals, e.g., hydroxyl (HO), peroxyl (RO_2_), superoxide anion (O^2−^), and alkoxyl (RO^−^) radicals, which disturb antioxidant defense mechanisms and induce chromosomal aberrations, gene mutations, and DNA damage [1,2,5,6]. To avoid plant cell damage by ROS, plants enhance various enzymatic and non-enzymatic antioxidant activities to avoid plasma membrane and other cell membrane damage by oxidative molecules [7]. Studies reported that a higher accumulation of glutathione reductase, catalase, ascorbate peroxidase, superoxide dismutase, and peroxidase enhances tolerance to plant Cd stress [8,9,10]. In the phenylpropanoid pathway, 10 enzymes are involved in catalysis including phenylalanine ammonialyase (PAL), peroxidase (POD), 4-coumarate CoA ligase (4CL), caffeic acid 3-O-methyl transferase (COMT), cinnamyl alcohol dehydrogenase (CAD), and cafeoy1-CoA3-O-methy1transferase (CCoAOMT) [11,12].

There are several mechanisms reported in the accumulation and translocation of Cd in the aerial part of plants. Cadmium accumulation in the plant system depends on physiological processes including binding on root cell walls, and sequestration in root vacuoles and xylem tissues [13]. The subcellular distribution of Cd showed that Cd mainly accumulated in the cell wall, followed by a soluble fraction in organelles and membrane in root, bark, and leaf tissues [14,15]. Cell walls serve as the first stress barrier consisting of pectin components. The higher amount of negatively charged groups in the pectin components positively interact with Cd, increasing the accumulation of Cd in the cell wall [16]. The soluble fraction was higher in the root system compared to the cell wall, organelle, and membrane fractions of bark and leaves [15]. The vacuole is the major cell organelle in plant cells and plays an important role in the retention of Cd. The metal-phytochelatin complex is sequestered in the vacuole and the mechanism is regulated by two cassette transporters (*AtABCC1* and *AtABCC2*) [17]. In addition, higher Cd accumulation and tolerance depend on the activation of several key genes associated with metal transporters, chelator proteins, antioxidant enzymes, defending genes, and transcription factors in plants [18]. Cadmium can move from the soil and accumulates in the edible parts of plants mediated by various transporters. Many transporter families play an important role in Ca^2+^, Fe^2+^, Mn^2+^, and Zn^2+^ absorption into plant cells and this may be involved in the transport of Cd. Nonessential elements (e.g., Cd) compete with other essential elements (Ca^2+^, Fe^2+^, Mn^2+^, and Zn^2+^) in the same membrane transport channels [18]. These transporter families include heavy metal ATPase (HMA), also known as P-type ATPases that absorb and transport both essential (Cu^2+^, Zn^2+^) and non-essential heavy metal ions (Cd^2+^ and Pb^2+^) [19]. Similarly, natural resistance-associated macrophage proteins (NRAMPs) uptake the Cd^2+^ ions [20], and iron-regulated transporter (*IRT1*) (ZRT/IRT-like proteins) are one of the ZIP family members of Cd^2+^ transport [21]. Moreover, cation exchangers (CAX) help to transport the Cd-chelate complex crossing tonoplast [13]. Furthermore, the SNF1-related protein kinase 2 subfamily protein (SnRK) is involved in environmental Cd stress signaling [22,23]. In addition, transcription factors (TFs) including MYB, WRKY, C2H2, bZIP, AP2, ERF, and DREB also play a significant role in metal stress tolerance in various plants by regulation of functional gene expression [24,25,26].

High throughput sequencing techniques provided fundamental knowledge of genes associated with biotic and abiotic stress in the plant system [24]. Many molecular approaches have been developed, among them, the RNA sequencing (RNA-seq) strategy, that plays an essential role and helps to identify expected and unexpected gene expression and regulatory networks including the pathogenesis mechanism of avocado [27], dwarfing regulation in Seashore paspalum [12], storage root development in sweet potato [28], drought-resistant mechanism of sorghum [29], and Cd stress in *Nicotiana tabacum* and *Nicotiana rustica* [13]. Seashore paspalum (*Paspalum vaginatum*), a halophytic warm-season turfgrass widely cultivated in salinity-affected areas and is applied for phytoremediation in Cd contaminated areas due to its excellent tolerance to salinity and Cd [12,30,31,32]. However, the molecular mechanism of Cd tolerance of Seashore paspalum remains unclear. In this study, we uncovered the molecular pathways of seashore paspalum response to Cd stress conditions using comprehensive RNA seq analysis. This finding will help to develop candidate Cd-tolerant genes for molecular breeding of Cd-tolerant turfgrass and other grasses.

## 2. Results

### 2.1. Antioxidant Activities in Paspalum vaginatum under Cd Stress

The enzymatic and non-enzymatic antioxidant enzyme activities were investigated under cadmium (Cd) stress conditions with different sampling intervals. All antioxidant activities were higher in the Cd1 sampling time compared to the Cd0 sampling time, but no significant difference was noticed between these timing intervals (Figure 1). Peroxidase (POD) and superoxide dismutase (SOD) activities were higher at 4 h sampling intervals compared to the 0 h and 1 h sampling intervals, but there was no significant difference in SOD activity compared to the 1 h sampling time (Figure 1A–C). Similarly, glutathione S-transferase (GST) activity was also significantly higher in the 24 h time point compared to 4, 1, and 0 h. However, GST was higher at the 4 h sampling intervals compared to 0 and 1 h sampling intervals, but there was no significant difference between the 4 h and 1 h sampling intervals (Figure 1D). The results showed that the activities of catalase (CAT), POD, and SOD were significantly higher at the 24 h sampling interval compared to other sampling times.

### 2.2. Transcriptomic Analysis after Cd Treatment

To understand the in-depth underlying molecular mechanisms of Cd stress tolerance of, RNA-sequencing of the Cd treated leaves was performed. A total of 12 cDNA libraries prepared from the leaves at 0, 1, 4, and 24 h after Cd treatment with three biological replicates from each timing sampled were used to identify the genes responsible for heavy metal tolerance in turfgrass. After assessing the quality and filtering the data, 84.91 gigabytes (Gb) were obtained from the samples. Each sample had an average of approximately 7.08 Gb of clean data. Of the 12 libraries, the GC% of the sequenced data varied from 51.19 to 52.29%, while more than 90.57% of the reads had an average quality score exceeding 30. The obtained high-quality sequencing results illustrated the suitability of subsequent analysis. The expression profile of DEGs was identified with false discovery rate (FDR)/*p* < 0.05 and fold change > 2 for comparison of Cd treatment with different sampling intervals and compared with different sampling intervals based on the FPKM value. Based on RNA-sequence data, DEGs were classified according to up-and down-regulation of genes as well as different time points after Cd treatment of seedlings.

A total of 6820 (4457 up-regulated, 2363 down-regulated) genes were expressed differentially between Cd1 and Cd0, while this value was 14,038 (9894 up-regulated, 4144 down-regulated) and 17,327 (7956 up-regulated, 9371 down-regulated) in Cd4 vs. Cd0 and Cd24 vs. Cd0, respectively. Similarly, 10,980 (7468 up-regulated, 3512 down-regulated) genes expressed differentially between Cd4 and Cd1, while 17,304 (6573 up-regulated, 10,731 down-regulated) and 11,121 (2658 up-regulated, 8463 down-regulated) genes expressed differentially between Cd24 vs. Cd1 and Cd24 vs. Cd4, respectively (Figure 2A). Furthermore, Venn diagram analysis displayed that among significantly regulated DEGs, 2170 genes were commonly regulated in all libraries, of which 796 DEGs were inclusively up-regulated and 1374 were down-regulated among all treatments, respectively. These results also showed that DEGs were higher in Cd24 vs. Cd0 (3227/5741, up-/down-regulated) followed by Cd4 vs. Cd0 (3743/1011, up-/down-regulated) and Cd1 vs. Cd0 (1857/376, up-/down-regulated) (Figure 2B,C).

### 2.3. Functions and Processes Influenced by the Cd Treatment

To further understand the modulated expression of the biological function of DEGs in *P. vaginatum* under Cd stress, DEG enrichment by GO analysis was performed, and the functions of the up/down-regulated DEGs were categorized. The GO enrichment of down-regulated DEGs showed that the most significantly enriched GO terms involved in the biological process were translation, protein metabolic process, generation of precursor metabolites, and energy biosynthetic processes. Of the cellular components, the cytosol, ribosome, and cytoplasm were highly enriched. In addition, molecular function category GO enrichment analysis showed that they are mostly involved in structural molecule activity, RNA binding, translation activity, and hydrolase activity (Figure 3A). Furthermore, to understand the biological functions and signal transduction pathways of these DEGs, we applied the KEGG pathway database to know the DEG-associated pathways under Cd stress. The highly enriched pathways were Valine, leucine, and isoleucine degradation (ko00280, 28), DNA repair and recombination proteins (ko03400, 175), genetic information processing (ko09120, 193), glycerophospholipid metabolism (ko00564, 25), and transcription machinery (ko03021, 182) (Figure 3B, Appendix A). For up-regulated DEGs, these enriched genes were annotated into three fundamental GO categories, namely, biological, cellular components, and molecular function. Of these GO terms, the importantly enriched GO terms involved in the biological process were response to chemicals, response to abiotic stimulus, and response to various stresses. Likely, the plasma membrane, membrane, vacuole, and peroxisome were higher enriched GO terms in the cellular component category. In addition, molecular function category GO enrichment analysis showed that they are mostly associated with DNA-binding transcription factor activity, transporter activity, and binding activity (Figure 3C). The highest enriched KEGG pathways were Transfer RNA biogenesis (ko03016, 15), C5-Branched dibasic acid metabolism (ko00660, 9), and Valine, leucine, and isoleucine degradation (ko00280, 11) (Figure 3D, Appendix A). Importantly, the obtained GO analysis and KEGG pathway enrichment analysis showed that the DEGs participated in many significant pathways in response to Cd stress. The response to abiotic stimulus, the metal transport mechanism, glutathione metabolism, and the transcription factor activity were among the most enriched pathways.

### 2.4. Cd Treatment Effects on Genes Involved in Glutathione Metabolism, Metal Transport, and Transcription Factors (TFs)

In the present study, we found that among the DEGs across the treatment groups, glutathione metabolism and phenylpropanoid biosynthesis pathways were significantly enriched under Cd stress conditions. In the glutathione metabolic pathway, there were 58, 111, and 82 genes up-regulated in the Cd1 vs. Cd0, Cd4 vs. Cd0, and Cd24 vs. Cd0 treatment samples, respectively. Among them, 11 genes were comparatively highly up-regulated between treatments. The up-regulated genes belonging to *glutathione synthase* (*GS*) (1 *GS*), *glutathione S-transferase* (*GST*) (7 *GST*), *glutathione peroxidase* (*GPX*) (2 *GPX*) and *glutathione hydrolase* (*GGT*) (1 *GGT*), which are the key enzymes of glutathione metabolism (Figure 4A).

Furthermore, a total of 54 metal transport genes were identified from the differentially expressed genes, and the number of DEGs among the treatments were significantly varied. The results showed that the treatment with Cd24 vs. Cd0 had more metal transporter-related DEGs than Cd4 vs. Cd0 and Cd1 vs. Cd0 treatments (Figure 4B. *NRAMP*, *HMA*, *ZIP*, and *CAX* were the significant enriched metal transporter genes detected in this study. In addition, five *HMA* genes (4/1, up-/down-regulated) were commonly enriched between the Cd1 vs. Cd0, Cd4 vs. Cd0, and Cd24 vs. Cd0 treatments (Figure 4C). Furthermore, *NRAMP* (2), *HMA* (13) (12/1, up-/down-regulated), *ZIP* (7) (5/2, up-/down-regulated), and *CAX* (1) genes were significantly enriched in Cd24 vs. Cd0 sampled plants. Likewise, *NRAMP* (2), *HMA* (10), *ZIP* (5), and *CAX* (1) genes were enriched in Cd4 vs. Cd0 treatments, *NRAMP* (0), *HMA* (5), *ZIP* (0), and *CAX* (0) genes were highly enriched in Cd1 vs. Cd0 (Figure 4B). Additionally, we also identified 6 *SnRK2* genes (6/1, up-/down-regulated) that were significantly enriched in the treatment of Cd24 vs. Cd0, and 3 *SnRK2* genes were positively enriched in Cd4 vs. Cd0, whereas there were no DEGs regarding *SnRK2* observed in the treatment of Cd1 vs. Cd0. These results indicate that higher metal transport-associated gene expression occurred with an increase of sampling time after Cd treatment.

The role of several Cd stress responsive transcription factors such as, MYB, AP2/ERF, WRKY, bHLH, etc., have been identified in different plants. In this study, several DEGs belonging to different transcription families (at least 65) such as ZIP, bHLH, MYB, AP2/ERF-ERF, Tify, C2H2, and WRKY were identified in *Paspalum vaginatum* under Cd stress. A total of 141 (up-/down-regulated), 425 (up-/down-regulated), and 426 (up-/down-regulated) differentially expressed genes associated with TF families were identified from the sample of Cd1 vs. Cd0, Cd4 vs. Cd0, and Cd24 vs. Cd0, respectively, under Cd stress Similarly, 366 DEGs encoding TFs were up-regulated in Cd4 vs. Cd0 followed by Cd24 vs. Cd0 (320 DEGs) and Cd1 vs. Cd0 (122 DEGs) treatments. In total, we also detected 52 DEGs that were commonly regulated encoding TFs between the Cd1 vs. Cd0, Cd4 vs. Cd0, and Cd24 vs. Cd0 treatments. As shown in Figure 4B, 13 TF families were identified from the predicted DEGs. Among these identified TFs, *AP2/ERF-ERF* and *Tify* are the largest TF families and each TF family has eight up-regulated DEGs followed by *C2H2* containing seven up-regulated DEGs. In addition, six up-regulated DEGs were found in the *WRKY* family, followed by four up-regulated DEGs identified both in the *MYB* and *NAC* families, three up-regulated DEGs in the *MYB-related* family; the *HB-HD-ZIP*, HSF, and *bZIP* family each contained two up-regulated DEGs, and finally, the *B3-ARF*, *E2F-DP*, *GARP-G2-like*, *MADS-M-type*, and *MADS-MIKC* TF families each contained only one up-regulated DEG. The results indicated that Cd stress in the plants induced higher expression of *AP2/ERF-ERF*, *Tify*, *C2H2*, and *WRKY* TF families.

### 2.5. Gene Expression Validation by qRT-PCR

For the validation of the expression trends of DEGs from RNA-seq data, we selected 4 highly expressed DEGs for qRT-PCR analysis and these genes were highly associated with metal transporter genes. The transcript abundance results showed that *CAX2*, *NRAMP2*, *HMA5*, and *SnKR2.7* were significantly induced by Cd stress, and the transcript abundance pattern of these genes were up-regulated at the time intervals. The highest gene expression was observed in 24 h after Cd stress (Figure 5).

### 2.6. Generation of PvSnRK2.7 Transgenic A. thaliana and Cd-Tolerance Analysis

To investigate the role of PvSnRK2.7 in Cd tolerance, two PvSnRK2.7 overexpression Arabidopsis lines (Line1 and Line2) were generated. Then, the effect of *PvSnRK2.7* overexpression on root growth was analyzed by observing Arabidopsis seedlings grown on 1/2 MS media with or without Cd stress (Figure 6). Genomic DNA and RNA (reverse transcription to obtain cDNA) were extracted from the leaves of T_2_ generation seedlings. *PvSnRK2.7* gene primers were designed for PCR and RT-PCR identification. The roots of all the plants were apparently similar under normal growth conditions, but the root growth of the WT plants were shorter under 80 µM CdCl_2_ than those under 0 µM CdCl_2_. Moreover, WT and two transgenic lines displayed similar root lengths under control conditions, whereas the root length of the WT plant was significantly shorter than transgenic lines under Cd stress. In addition, the fresh weight of the roots of transgenic plant roots were significantly higher than WT plants under normal as well as Cd stress conditions. These results indicated that overexpression of *PvSnRK2.7* promotes root growth under normal and Cd stress conditions.

### 2.7. Antioxidant Enzyme Activities Are Increased in PvSnRK2.7 Overexpressing Plants

Several studies referred to the protective role of antioxidant enzyme activities in plants under Cd stress. To investigate the antioxidant enzyme activities after Cd exposure, we measured antioxidant enzyme activities level in PvSnRK2.7-Arabidopsis and control plants. In the absence of Cd, similar levels of superoxide and hydrogen peroxide were produced in both PvSnRK2.7 transgenic and control plants. In the presence of 80 μM Cd, PvSnRK2.7 transgenic plants exhibited a significantly higher induction in the CAT, POD, SOD, and GST activities, compared with control plants. Increases in CAT, POD, and SOD activities were rather small. Only GST activity clearly increased in both transgenic lines after 14 d of treatment (Figure 7).

## 3. Discussion

Plants have evolved various effective strategies to manage heavy metal toxicity, including Cd, but the molecular mechanism of plants exposed to Cd stress and accumulation in plant systems is still poorly understood [10]. Primarily under stress conditions, the physiological process of plant cells was altered, thus increasing reactive oxygen species, leading to a series of defense reactions. To mitigate the cell damage caused by ROS, the plant activates both the enzymatic and non-antioxidant enzyme system, which promotes plant stress tolerance by removing or regulating excess ROS production and avoiding oxidative cell damage [24]. Under metal stress conditions, the plant activates various antioxidant enzymes, among which SOD, POD, CAT, GSH, and GST are the main antioxidant enzymes playing a crucial role in stress tolerance [33]. In this study, we observed that the enzymatic antioxidants (CAT, POD, and SOD) and non-enzymatic antioxidant (glutathione S-transferase/GST) activity was higher in different sampling intervals under Cd stress conditions (Figure 1). However, a significant level of antioxidant activity was observed in the 24 h sampling points followed by others (0, 1, and 4 h). Similar results were observed in other research investigations, and they concluded that Cd stress increased POD, CAT, SOD, and GST activities, and proline content in Populus × canadensis and tartary buckwheat [34]. SOD is the first antioxidant enzyme that plays a significant role in ROS metabolism and increase H_2_O_2_ production by dismuting of O_2_•−/HO_2_•−. Subsequently, higher accumulation of H_2_O_2_ is reduced by CAT and POD, which helps to reduce oxidative cell damage caused by excess H_2_O_2_ [35]. Previous studies also have reported that GST, GPX, and reduced glutathione (GSH) regulate ROS production under various stress conditions including drought, fungal infection, and Cd stress [29,36,37].

To elucidate the Cd-stress induced inherent mechanism at the transcriptomic level, we extensively conducted the present study in *Paspalum vaginatum*. GSH metabolism is significantly involved in metal tolerance and accumulation [38]. Here, we identified several glutathione (GSH) metabolism related enriched DEGs such as GS, GST, GPX, and GGT genes in Cd stressed plants (Figure 4). Similar to our findings, we also observed that the GSH and GST genes were up-regulated in *Populus × canadensis* under Cd stress. In addition, GSH was specifically allocated to phytochelatin synthesis (PC), which chelates heavy metal ions, including Cd^2+^ [39]. The higher expression of GS, GST, and GPX in this study suggested that they may be significantly involved in both plant defense and metal chelation of Cd. Lignin acts as a physical barrier that helps immobilize Cd into the secondary cell wall and prevents the absorption of Cd into the protoplast [11]. In the present study, we identified five genes; namely, peroxidase, cytochrome P450, 4CL, O-methyltransferase (OMT), and beta-glucosidase in phenylpropanoid biosynthesis which are associated with the enrichment of lignin synthesis in *Paspalum vaginatum* under Cd stress [13]. We observed that the 4CL genes were significantly enriched in *Nicotiana tabacum* and *Nicotiana rustica* under Cd conditions. We also noted higher accumulation of Cd in *N. tabacum* xylem sap compared to *N. rustica* and this might be related to higher expression of 4CL and casparian strip genes in the *N. rustica* plant species thus helping to reduce the accumulation of Cd in *N. rustica*. Similarly, The higher lignin content increases the plant stability and robustness against mechanical, chemical, and environmental damage such as wounding, heavy metals, and drought [40], which is strongly supported by our present findings.

The vacuole is the largest cell organelle in mature plant cells, and metal toxic substances (Cd and arsenic) are stored in this organelle [38]. Ion transporters help with uptake and translocation of essential elements in plants such as ABC families, HMA, NRAMP, and ZIP [13,26]. In this study, we identified 23 transporter genes belonging to HMA, NRAMP, ZIP, and CAX, and they were differentially expressed at different sampling intervals. The maximum number of DEGs were observed in Cd24 vs. Cd0 followed by Cd4 vs. Cd0 and Cd1 vs. Cd0; indicating that different gene expression regulation occurred based on various time intervals between treatments. In addition, we also identified 13 HMA proteins encoding genes and 12 HMA genes (HMA2/HMA4/HMA5) that were significantly enriched in *Paspalum vaginatum* under Cd stress and the qRT-PCR results (HMA5) confirmed this upregulation (Figure 4). Higher expression of HMA2/HMA4 in *N. tabacum* indicates a stronger translocation of Cd from the root to the shoot of the plants [24], which strongly supports our present findings. The ZIP protein is the best nonspecific transporter and helps the uptake of Cd as Cd+ into the plant system [24,41]. In the current study, we identified seven genes involved in ZIP transport, among which four genes were highly enriched in *Paspalum vaginatum* exposed to Cd stress; suggesting a higher accumulation of Cd^2+^ in plant leaves. CAX and NRAMP ion transporters also play a significant role in the translocation and compartmentalization of Cd into the vacuole of plant cell vacuole cells [38]. In this study, higher expression CAX2 and NRAMP2 genes was observed in *Paspalum vaginatum* under Cd stress (Figure 5). The enriching activities of SaCAX2h in *Sedum alfredii* was increased Cd accumulation in tobacco plants [42], and NRAMP1, NRAMP3, and NRAMP4 are involved in Cd^2+^ accumulation in Arabidopsis and rice plants, while the NRAMP2 genes can transport Mn^2+^ and Fe^2+^ ions [38,43,44], our findings were consistent with previous results.

Transcription factors (TFs) play a significant role in stress tolerance in plants [24]. In the present study, we identified 65 TF families including bZIP, bHLH, MYB, NAC, HSF, Tify, AP2/ERF-ERF, and C_2_H_2_, and, among them, AP2/ERF-ERF, Tify, and C_2_H_2_ are the largest TF families that were significantly enriched under Cd stress (Figure 4). Our findings were strongly supported by previous research that identified a total of 59 TF families (AP2-ERBP, C_2_H_2_, NAC, MYB, and MYB-related are the most prominent families) that are highly expressed in two barley genotypes under copper and cobalt stress conditions [26]. In addition, Tify, previously known as ZIM, is also one of the important members of TF families playing a significant role in the cross-talk between phytohormones and their signaling pathways, which are involved in biotic and abiotic stresses [45]. In this study, we observed a higher number of genes in the Tify TF family genes under Cd stress (Figure 4). Similar results were also found by the Wang et al.’s [46] research team and they identified 25 TIFY genes in the Populus trichocarpa genome. Moreover, C_2_H_2_ (Cys2/His2-type) zinc finger proteins are key transcriptional regulators in response to various abiotic stresses including extreme temperatures, salinity, drought, oxidative stress, excessive light, and silique shattering [47]. The higher number of C_2_H_2_ TF families associated with gene expression was identified under Cd stress (Figure 4) and analogous results were observed by Lwalaba et al. (2021) [26] in barley under cobalt and copper stress. Furthermore, we identified some other common TF families such as WRKY, MYB, NAC, MYB-related, HB-HD-ZIP, HSF, B3-ARF, E2F-DP, GARP-G2-like, MADS-M-type, and MADS-MIKC that were up-regulated in plants under Cd stress. Previous studies have reported that WRKY, MYB, and NAC are the predominant transcription factors playing a significant role in plant metal stress tolerance mechanisms [24,26].

Sucrose non-fermenting (SNF1) related protein kinase 1 subfamily protein (SnRK) plays a crucial role in ABA-dependent and ABA-independent environmental stress-signaling responses [48,49]. In this study, we identified that SnRK2.7 was up-regulated in *Paspalum vaginatum* under Cd stress and plants overexpressing SnRK2.7 could enhance tolerance to Cd (Figure 5 and Figure 6). SnRK2 is a relatively large gene family and is significantly expressed in both cells and seedlings of *Arabidopsis thaliana* [50] and *Oryza sativa* [51] under different stress conditions. Similarly, [52] reported that ABA signaling pathways participated in Cd detoxification in the low-Cd-accumulating genotype (LAJK) of *B. chinensis* by ABA-induced antioxidant pathway. Based on these study results, we concluded that the enriched expression of the SnRK2 genes in *Paspalum vaginatum* under Cd stress conditions indicate the active involvement of ABA signaling in the regulation of heavy metal stress regulation in plants. Finally, we concluded that antioxidant activities and their encoding gene expression, stress-responsive genes, metal transporter genes, stress signaling response genes, and transcription factor regulation-associated genes were highly enriched with increasing sampling time points compared with earlier sampling time points and thus all help to increase Cd tolerance of the plant.

## 4. Materials and Methods

### 4.1. Plant Materials and Cd Treatment

*Paspalum vaginatum* SW (Seashore paspalum) were grown in 11-cm diameter and 21-cm deep plastic pots filled with 1:1:1 ratio of sand, soil, and perlite. The experimental pots were placed in turfgrass germplasm (Nanjing Agricultural University, Jiangsu province, China) and kept at optimal temperature and relative humidity. An adequate level of water and fertilizer was supplied to the plants. The plants were treated with 300 µmol/L of Cd^2+^, supplemented with CdCl_2_⋅5H_2_O. After Cd treatment, *Paspalum vaginatum* leaves were harvested at 0, 1, 4 and 24 h time intervals, immediately freeze-dried, and stored at −80 °C for transcriptome analysis, gene expression validation, and antioxidant analysis.

### 4.2. Antioxidants Activities Estimation

Leaf samples were collected at 0, 1, 4, and 24 h after Cd treatment and then processed for the analysis of antioxidant enzyme activities as described [53]. Catalase (CAT) activity was determined by following the consumption of H_2_O_2_ at 240 nm for 3 min [54], peroxidase (POD) activity was measured by the change in absorbance of 470 nm due to that determined using 4-methyl catechol as substrate according to the method described [55]. Superoxide dismutase (SOD) activity was determined by measuring the inhibition in the photoreduction of nitroblue tetrazolium (NBT). Glutathione S-transferase (GST) activities were determined according to the manufacturer’s instructions (Jiancheng Bioengineering Institute, Nanjing, China). The antioxidant enzyme activities were statistically analyzed with three independent replicates.

### 4.3. Transcriptome Assembly and Genes Annotation

The raw data were filtered, and then clean data were obtained by removing adapter sequences, poly-N, and low-quality reads. Simultaneously, GC content levels, Q20, and Q30 of the clean data were evaluated. The functions of the unigenes were annotated using a series of databases, including BLASTx, against NCBI non-redundant protein (Nr), NCBI nucleotide collection (Nt), and Swiss-Prot databases. The reference genome was directly downloaded from the genome website (https://www.ncbi.nlm.nih.gov/genome/?term=Paspalum+vaginatum (accessed on 14 July 2022)). After filtering, the reference genome was built using hisat2-build and paired-end clean reads were aligned to the reference genome using hisat2 v2.0.12 software (Johns Hopkins University, Baltimore, MD, USA). Functional annotations of putative unigenes were grouped using the Kyoto Encyclopedia of Genes and Genomes (KEGG) Gene Ontology (GO).

### 4.4. Gene Expression Level Quantification and Differential Expression Analysis

Gene expression levels were calculated using the fragments per kilobase of transcript sequence per million base pairs sequenced (FPKM) method. Differentially expressed genes (DEGs) were identified via differentially DEseq package [54] The DEGs were identified with false discovery rate (FDR)/fold change > 2 and *p* < 0.05 were deemed to be DEGs. Gene Ontology (GO) enrichment analysis of DEGs was carried out by the GO seq R package. GO terms with corrected *p* < 0.05 were significantly enriched.

### 4.5. RNA Extraction, cDNA Library, and qRT-PCR for Validation

To evaluate the reliability of differentially expressed genes under Cd stress conditions revealed by RNA sequencing, four genes were selected from the important pathways and validated by quantitative real-time RT-PCR (qPCR). The leaf samples were collected from Cd-treated plants at 0, 1, 4, and 24 h intervals, and freeze-dried in liquid nitrogen. Total RNA was extracted from the ground leaves according to the manufacturer’s instructions (EASYspin RNA Plant Mini Kit, Aidlab, China). Extracted total RNA was reverse-transcribed to cDNA in a total volume of 10 μL using the TransScript II cDNA synthesis SuperMix kit (Transgen Biotech, China). The primer sequences were designed based on the corresponding gene sequence by NCBI (https://www.ncbi.nlm.nih.gov/tools/primer-blast/, accessed on 14 July 2022). Gene expression analysis was performed using a qRT-PCR CFX96 thermocycler (Bio-Rad) with GAPDH serving as an internal control. The primers of target genes and GAPDH were diluted in a ChamQ SYBR qPCR master mix (Vazyme Biotech Co., Ltd., Nanjing, China), and the mix was transferred to a 96-well plate. Reactions were performed and relative expression values were calculated according to the 2^−ΔΔCt^ [55].

### 4.6. Statistical Analysis

GraphPad Prism 5 (GraphPad software, San Diego, CA, USA) was used to compute the data and statistical analyses. All the records were statistically analyzed with appropriate replications, the values are expressed as the mean value with standard error means (SE±), and the results were statistically analyzed using one-way ANOVA (analysis of variance) with a Tukey’s correction (*p* < 0.05).

### 4.7. Agrobacterium-Mediated Transformation of Arabidopsis

The monoclonal colonies containing the target gene were inoculated into 1 mL culture medium (50 mg/L Kan and 50 mg/L Rif), and shaken at 28 °C for 2–3 days, then 100 μL of solution was extracted and inoculated into 200 mL of fresh medium (OD600 = 0.2) allowing growth until OD600 = 0.8 under the same conditions. After centrifugation at 4 °C, the thalli were collected in a 50 mL tube. Next, the thalli were resuspended with pre-cooled 5% sucrose solution until no obvious thalli ware found. Before dipping, 1–2 drops of Silwet-L77 surfactant were added. Apical parts of the Arabidopsis (growing to 2 months old) were dipped in the dipping dye for about 5 s and shaken slightly.

### 4.8. Positive Lines Screening of Transgenic Arabidopsis

The primary seeds were harvested after *Agrobacterium*-mediated transformation was recorded as the T_0_ generation. The T_0_ generation seeds were sown on 20 mg/mL Basta resistance screening 1/2 MS culture medium. After 7 days, the normal surviving seedlings were transplanted into the soil and cultured until maturity, and the harvested seeds were recorded as the T_1_ generation. According to the above methods, until the T2 generation seeds were obtained, gDNA and RNA (reverse transcription to obtain cDNA) were extracted from the leaves of the T_2_ generation seedlings. Precise gene primers were designed for PCR and RT-PCR identification.

### 4.9. The Treatment of Cadmium on Culture Medium

The transgenic *Arabidopsis* seeds of the T_2_ generation were disinfected (20% 84 disinfectant for 20 min, washed 6 times with sterile water, and vernalized at 4 °C for 2 days) and then seeded on 1/2 MS medium. After germination, seedlings with consistent growth were selected and transferred to a new square medium containing different concentrations of cadmium. Seedlings grew under light (photoperiod 16 h/8 h) until the obvious phenotype was observed, then pictures were collected and root length and fresh weight were measured. Three biological replicates were performed for each treatment, collected at 0, 1, 7, and 14 days after Cd treatment, and then processed for the analysis of antioxidant enzyme activities as described in Section 4.2.

## 5. Conclusions

The transcriptomic profile obtained from the results of the RNA seq analysis revealed a distinctly different gene expression pattern between the sampling intervals after Cd treatment in *Paspalum vaginatum*. A higher number of DEGs were observed in Cd24 vs. Cd0 (17,327 DEGs), followed by Cd4 vs. Cd0 (14,038 DEGs) and Cd1 vs. Cd0 (6820). The DEG results indicated that higher Cd accumulation and stress response occurred with increasing sampling times. The schematic diagram represents the transcriptional changes of genes responsible for Cd stress tolerance and accumulation in *Paspalum vaginatum*. In this study many genes encoding lignin biosynthesis (POD, 4CL, and OMT), non-enzymatic antioxidants (GST, GS, and GPX), signaling response (SnRK2), and transcription factors (AP2/ERF-ERF, Tify, C2H2, and WRKY) were highly up-regulated in the plants under Cd treatment. Similarly, heavy metal transporter genes (HMA, ZIP, CAX2, and NRAMP2) were highly expressed among the treatments (Cd1 vs. Cd0, Cd4 vs. Cd0, and Cd24 vs. Cd0) and their expression increased with an increase of sampling time indicating that higher Cd accumulation occurred based on the incubation intervals. Based on transcriptome analysis, *Paspalum vaginatum* can be used as a heavy metal remover from polluted soil, however, further experimental studies are required to conclude this statement.

## Figures and Tables

**Figure 1 plants-11-02078-f001:**
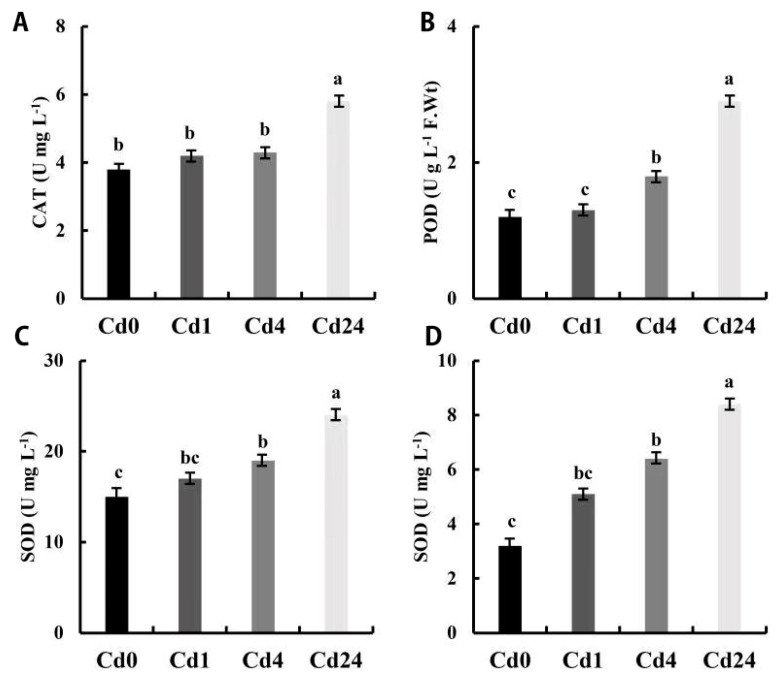
Enzymatic and non-enzymatic antioxidant activities in leaves of *Paspalum vaginatum* under stress from Cd. (**A**) Catalase (CAT); (**B**) peroxidase (POD); (**C**) superoxide dismutase (SOD); (**D**) glutathione S-transferase (GST). Different letters in the column indicate significant differences at *p* < 0.05, according to Tukey’s test; Vertical bars indicate standard errors of each mean value (n = 3).

**Figure 2 plants-11-02078-f002:**
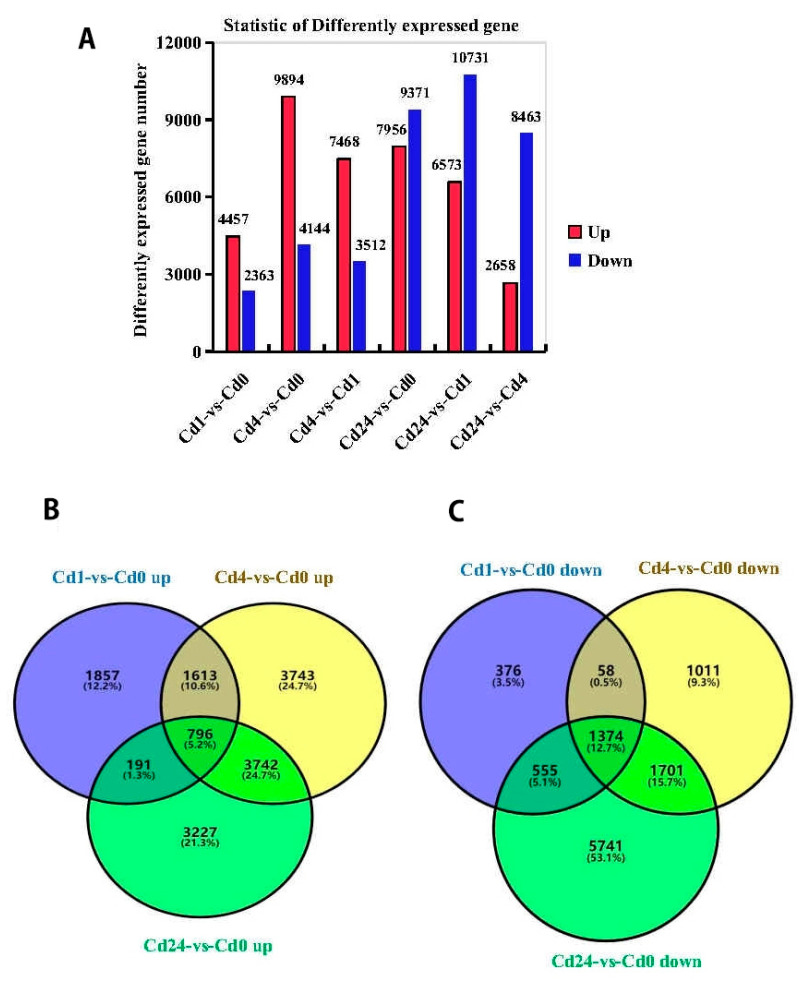
Analysis of gene expression by cadmium (Cd) stress tolerance of *Paspalum vaginatum*. (**A**) Compiled data of differentially expressed genes (DEGs) at different sampling times in *Paspalum vaginatum* under cd stress; (**B**) Venn diagrams of up-regulated DEGs for Cd1 vs. Cd0, Cd4 vs. Cd0 and Cd24 vs. Cd0. (**C**) Venn diagrams of down regulated DEGs for cd1 vs. Cd0, Cd4 vs. Cd0, and Cd24 vs. Cd0.

**Figure 3 plants-11-02078-f003:**
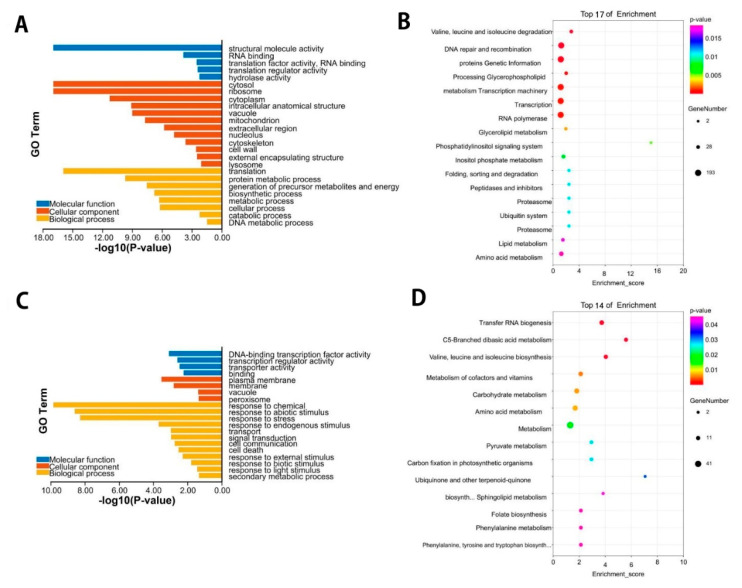
Gene ontology (GO) enrichment and KEGG enrichment analysis of DEGs. (**A**) Go terms of down-regulated DEGs in *Paspalum vaginatum* under Cd stress. (**B**) Graphs present 17 KEGG pathways with the highest transcriptional variations out of the down-regulated DEGs in *Paspalum vaginatum*. (**C**) Go terms of up-regulated DEGs in *Paspalum vaginatum* under Cd stress. (**D**) Graphs present 14 KEGG pathways with the highest transcriptional variations, out of the up-regulated DEGs in *Paspalum vaginatum*.

**Figure 4 plants-11-02078-f004:**
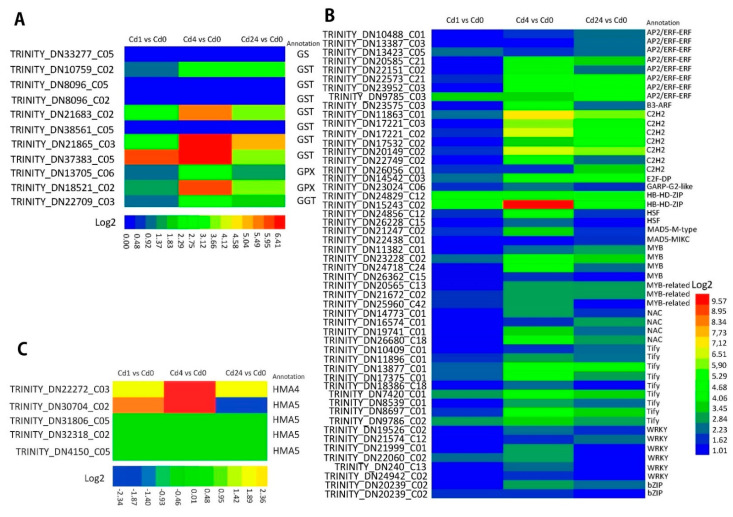
Heat map of DEGs involved in glutathione metabolism, metal transporter, and transcription factors. (**A**) Heat map of DEGs involved in glutathione metabolism; (**B**) Heat map of DEGs involved in metal transport; (**C**) Heat map of DEGs involved in transcription factors.

**Figure 5 plants-11-02078-f005:**
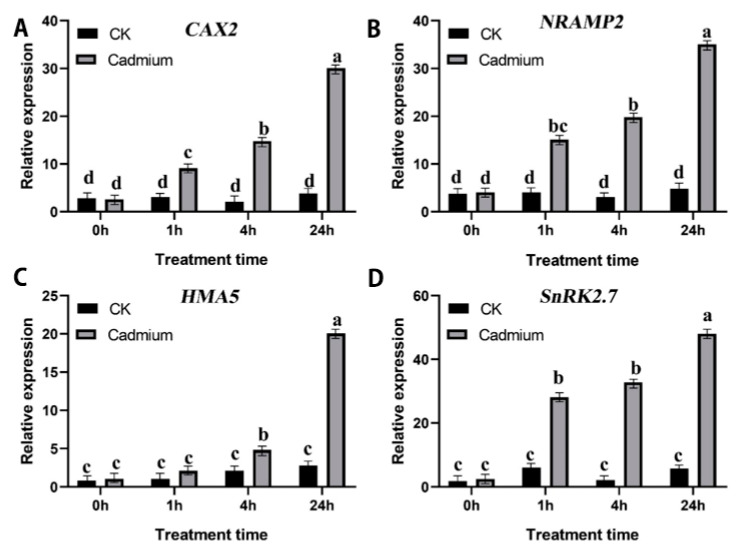
Validation of the expression pattern of 4 selected DEGs in the RNA-Seq by qRT-PCR in *Paspalum vaginatum* root after Cd treatments. (**A**) *CAX2*. (**B**) *NRAMP2*. (**C**) *HMA5*. (**D**) *SnKR2.7*. The lowercase letters above columns represent significant differences between over-expressed and control lines under different treatments (*p* < 0.05).

**Figure 6 plants-11-02078-f006:**
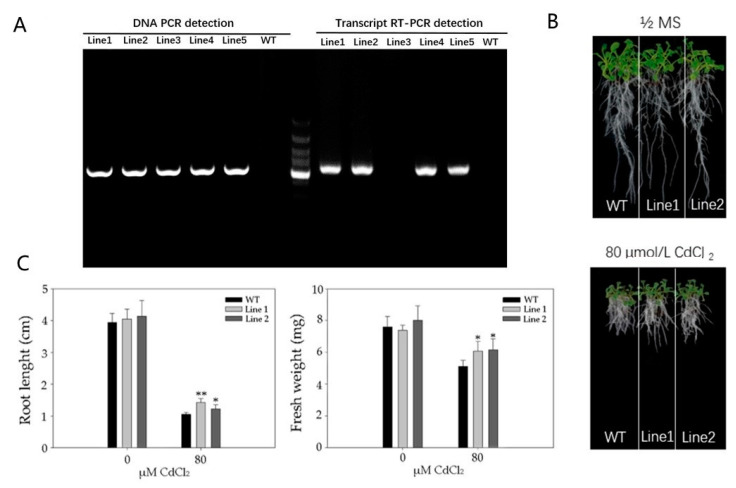
Cd resistant phenotypes of the *PvSnRK2.7* over-expression lines (Line-1 and Line-2). WT and PvSnRK2.7-OE seedlings were grown on 1/2 MS nutrient solution containing 0 or 80 μM CdCl_2_ for 14 d. (**A**) *PvSnRK2.7* gene primers were designed for PCR and RT-PCR identification, then phenotypes were photographed (**B**), the root length and fresh weight were measured (**C**). The mean value and standard error were obtained from 3 biological replicates, and the significance difference level *p* ≤ 0.05 (*), *p* ≤ 0.01 (**).

**Figure 7 plants-11-02078-f007:**
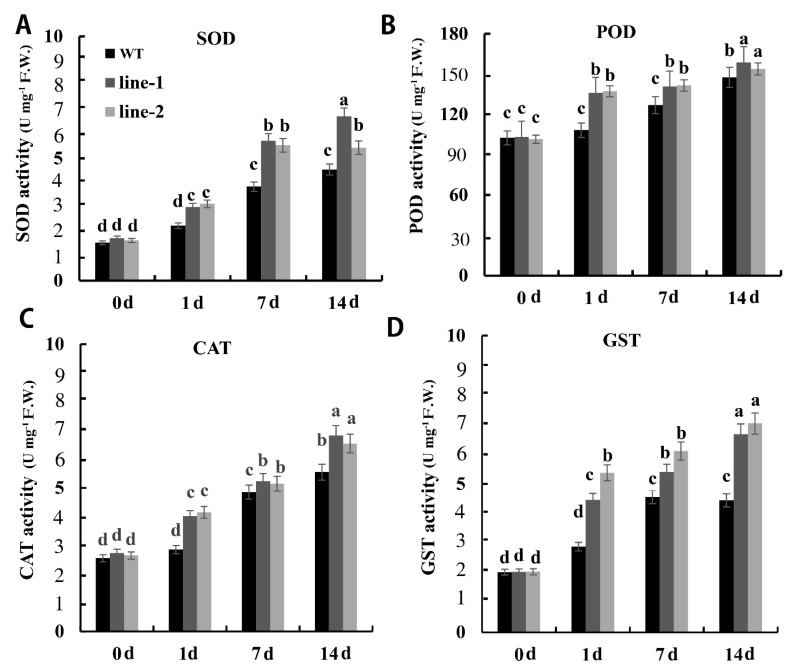
Oxidative stress in *PvSnRK2.7*-overexpressing and control Arabidopsis. (**A**) Superoxide, (**B**) hydrogen peroxide, (**C**) malondialdehyde and (**D**) glutathione S-transferases contents in *PvSnRK2.7*-expressed and control plants. Three-week-old seedlings grown on 1/2 MS media with and without 80 μM CdCl_2_ were used for experiments. Data indicate the mean ± SE of three independent biological experiments. The lowercase letters above columns represent significant differences between over-expressed and control lines under different treatments (*p* < 0.05).

## Data Availability

The datasets used in this study can be found in the NCBI SRA database under the accession number SUB11901595.

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
