# Peer review of "Transcriptome Analysis Reveals the Stress Tolerance to and Accumulation Mechanisms of Cadmium in Paspalum vaginatum Swartz"

_plants, 2022, doi:10.3390/plants11162078_

Round 1
Reviewer 1 Report
The manuscript describes a transcriptomic analysis of Paspalum vaginatum treated with 300 µM Cd grown in a mixed substrate. Leaves were collected from 0 to 24 h of exposure. At the same time, several redox enzymatic activities were analysed. Form the transcriptomic data a set of overexpressed genes were further validated using RT-PCR. Among these genes, a protein kinase (SnRK2.7), involved in ABA signally, was overexpressed in transgenic Arabidopsis plants, to check for its contribution in Cd tolerance. The authors used state-of-the-art techniques, but some experiments need improvement, and additional results are required to enhance the scientific value of the manuscript. These are the major problems found:
· CAT, SOD, POD and GST enzymatic activities were analysed as indexes of Cd-induced oxidative stress, which all eventually increased with time of Cd treatment. However, it is not clear that some of these enzymes were subject to transcriptional regulation, but apparently GST. Or I must say GST genes… due to the large family of GST genes existing in most organisms. I do not see the point of grouping the discussion of those redox enzymes (but GST) with GSH metabolism, based on the changes observed in several GSH-related genes. Perhaps additional data of GSH concentration, or phytochelatins accumulation may help with this discussion.
· The resolution of Gene ontology (GO) enrichment and KEGG enrichment analysis of DEGs (Fig. 3) and heat map Heat map of DEGs involved in glutathione metabolism, metal transporter and transcription factors (Fig. 4) are very low. It is difficult to visualise gene families and fold changes. Why some gene expression increases at certain times of treatment (Cd1 or Cd4) while decreased at longer times (Cd24)?
· Fig. 5 shows RT-PCR validation of expression detected for several genes of interest. Why SnRK2.7 expression was higher in Cd0 plants compared with CK0 plants? If Cd treatment was so short, why did it increased?
· Root length and fresh weight of transgenic Arabidopsis overexpressing PvSnRK2.7 were very similar to those of WT plants under 80 µM Cd stress. Differences were minuscule, considering the standard errors in Fig. 6. Probe must be given to show proper expression of the transgene (DNA insertion, mRNA levels, protein immnunodetection), which might explain the poor tolerance response found in both tested transgenic lines.
· Similarly, why Cd-responding redox and GST enzymatic activities were not measured in the Arabidopsis experiments? If those enzymes are good indexes of Cd responses must be tested also here.
· Provide details of enzymatic activity determination: How were them measured?
Minor points: some chemical formula are wrong (see ROS formula of superoxide anion O2-; Cd+ should be Cd2+). Why they used this section title?: "2.6. Generation of PvSnRK2.7 transgenic A. thaliana and Salt-tolerance analysis". 4.2. Antioxidants Estimation should be 4.2. Antioxidants analyses??
Author Response
Thank you very much for reading this manuscript and providing kind comments. We have revised the parts of the manuscript that you pointed out. The revision manuscript is shown in the attachment.
Q1、Answer: Thanks for your helpful suggestions. We have revised the discussion to point out the relationship between transcription regulation and those enzymatic activities. We think this is a very good question. Following transcriptome data, the expression level of several family genes (CAT, SOD, POD) was also upregulated in response to cadmium stress unshown in manuscript. In other plants, the accumulation of GSH and phytochelatins have been verified for improving cadmium tolerance. The relevant research in Paspalum vaginatum will be performed in future work.
Q2、Answer: Thanks for your helpful suggestions. According to the analysis of Go and KEGG (Fig. 3), there are many pathways in response to cadmium stress. Following the known reports in other plants, we focused on the changes of glutathione metabolism, metal transporter and transcription factors shown in heat map (Fig. 4). About gene expression patterns, many genes especially transcription factors exhibited an early rapid response in transcription level and then recovery to initial level at longer times in several researches. Additionally, in our onging work, although the transcription level of several genes restored the initial paper level at longer times, their protein level appeared the changes.
Q3、Answer: Thanks for your reminder. We checked the raw data and reanalyzed the results. Due to data entry error, the results appeared deviation. We have recalculated the data and found the expression level of SnRK2.7 expression in Cd0 plants and CK0 plants presenting no significant difference shown in revised Fig.5.
Q4、Answer: Thanks for your suggestions. Through phenotype and data detection, we can determine the better cadmium tolerance of PvSnRK2.7 overexpressing Arabidopsis comparing to wild type. We supplemented the photograph of PCR detection in DNA level and RT-PCR detection in mRNA level (Fig.6a), presenting the reliability of transgenic Arabidopsis.
Q5、Answer: Thanks for your helpful suggestions. We have supplemented the data of 4 antioxidant enzymes activities in Arabidopsis experiments (Fig.7). The results showed the overexpression of PvSnRK2.7 can improve the enzymes activities.
Q6、Answer: We have added the detailed information on enzymatic activity assays in Line 409-418 in Materials and methods.
Minor points: As your suggestions, we have revised those writing mistakes and rechecked the manuscript.
The manuscript has been reedited by a native English speaker who is familiar with the research field.

Reviewer 2 Report
I have gone through the manuscript entitled "Transcriptome analysis reveals the stress tolerance and accumu- 2 lation mechanisms to cadmium in Paspalum vaginatum Swartz". i have very minor questions as some more latest refrences should be added in the introduction part. and discussion part should be more elaborated.
also some typo errors should be rectify
Author Response
Thank you very much for reading this manuscript and providing kind comments.
As your suggestions, we added several latest references in introduction and discussion part. We also revised those writing mistakes and rechecked the manuscript. The revision manuscript is shown in the attachment.

Reviewer 3 Report
Minor corrections should be made to the manuscript:
Line 14: CAT, POD, SOD and GST, write the full names of the enzymes, not the acronyms
Line: 24 and other lines: Arabidopsis in italics
Line 380: -80 oC, missing Celsius degrees
Line 441: agrobacterium in italics: Agrobacterium and initial capital letter
Author Response
Thank you very much for reading this manuscript and providing kind comments.
As your suggestions, we revised those writing mistakes and rechecked the manuscript. The revision manuscript is shown in the attachment.

Round 2
Reviewer 1 Report
It is commented in line 282 that "PvSnRK2.7 transgenic plants exhibited a markedly higher induction in the CAT, POD, and SOD activities, compared with the control plants (Figure 7)." However, I think the it was just SIGNIFICANT. Increases in CAT, POD and SOD activities were rather small. Only GST activity clearly increased in both transgenic lines after 14 d of treatment.
In the text (line 186) is not mentioned the "Significantly enriched top 20 KEGG pathways of down-regulated DEGs in Paspalum vaginatum under Cd stress". Rephrase this "top-20" term, by something like "graphs present 20 KEGG pathways with the highest transcriptional variations, out of the XX includes in Supplementary data??"
Use italics in plant species names (example of line 329) "N. tabacum xylem sap compared to N. rustica and this might be related to higher expression of 4CL and casparian strip (suberization??) genes in the N. rustica plant species thus helping to reduce the accumulation of Cd in N. rustica."
Correct "glutathi-one" in line 312
Change the format of Fig. 1 to match that of Fig. 7. The resolution and type of bars is much better in Fig. 7.
The resolution of Figs. 2, 3 and 4 is very low. It must be improved substantially. It is still very difficult to read the names and values shown.
Author Response
Thank you very much for reading this manuscript and providing kind comments. We have revised the manuscript that you pointed out.
Q1:In line 282-286, we have revised the description of Fig.7 following the suggestions.
Q2: In line 186-190, we have revised the description of Fig.3 following the suggestions.
Q3: We have corrected the italics format of plant species names throughout the manuscript.
Q4: We have Correct "glutathi-one" .
Q5: We have revised the format of Fig.1.
Q6: We have revised it and improved the resolution of Fig.2-4.